# MedCalc-R1: Knowledge-Guided Reward Framework for Medical Mathematical Reasoning

## Abstract

Medical mathematical reasoning is a critical component of clinical decision-making, where accuracy directly affects patient safety and treatment outcomes. However, existing large language model approaches, despite improving complex reasoning, still suffer from knowledge degradation, computational bias, and limited interpretability. While reward mechanisms in large language models are commonly based on coarse-grained acceptable ranges, they fail to ensure stable and precise mathematical outputs. To address these challenges, we propose a knowledge-guided reward framework with two complementary mechanisms. First, a knowledge verification reward enforces explicit formula generation and leverages an independent verification model to check both formulas and results, thereby mitigating knowledge forgetting, enhancing interpretability and reasoning transparency. Second, a hybrid soft–hard reward mechanism incorporates clinical safety thresholds as hard constraints and introduces progressive accuracy-based rewards as soft optimization, simultaneously achieving improvements in both safety and precision. Extensive experiments on medical mathematical reasoning tasks demonstrate that our approach significantly outperforms existing methods in terms of reasoning accuracy, knowledge robustness, and model generalization, thereby validating the effectiveness and broad applicability of the proposed framework.

## 1 Introduction

Medical mathematical reasoning plays a central role in clinical decision-making, where accuracy directly determines patient safety and treatment outcomes (Singhal et al., 2023; Thirunavukarasu et al., 2023; Lucas et al., 2024). This is especially critical in high-risk scenarios such as calculating drug dosages based on creatinine clearance or performing individualized risk prediction through complex scoring formulas. Even minor computational errors in these contexts may result in inappropriate dosing, diagnostic bias, or life-threatening consequences (Cicero et al., 2020; Hijji, 2025). With the rapid development of artificial intelligence, large language models (LLMs) have demonstrated remarkable capabilities in complex reasoning tasks, offering new opportunities for medical decision support (Xu et al., 2025; He et al., 2025; Wang et al., 2025b). Nevertheless, the stringent requirements of clinical practice—demanding high accuracy, transparency, and safety—remain unmet, and the reliability of existing approaches in mathematical reasoning is still far from clinically acceptable standards.

In recent years, LLMs have achieved strong performance on general complex reasoning tasks such as mathematical reasoning (Shao et al., 2024; Yang et al., 2024b), logical inference (Yang et al., 2024c; Qin et al., 2024; Liu et al., 2025a), and program synthesis (Austin et al., 2021; Guo et al., 2024). In particular, reinforcement learning (RL) has been combined with LLMs to optimize reasoning trajectories through reward signals, leading to notable progress (Wang et al., 2024). For example, o1 (Jaech et al., 2024) and DeepSeek R1 (Guo et al., 2025) have shown impressive results on complex mathematics and long-chain reasoning tasks, highlighting the growing generalization ability of LLMs under numerical and logical constraints. These advances suggest promising opportunities for applying LLMs to high-stakes domains such as medical mathematical reasoning. Nevertheless, progress in this direction has been limited, as direct transfer to clinical scenarios exposes several

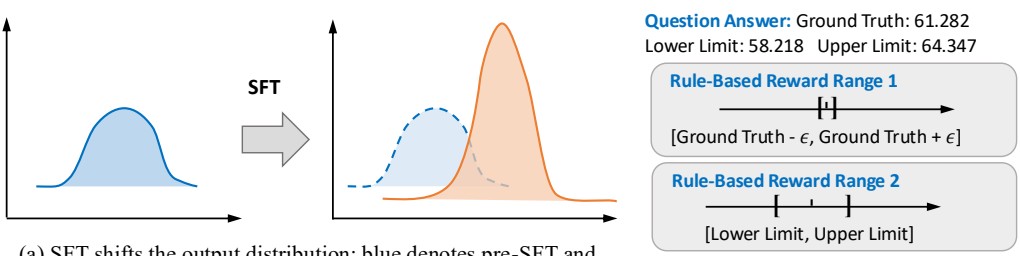

(a) SFT shifts the output distribution: blue denotes pre-SFT and orange post-SFT.

(b) Limitations of rule-based reward design.

Figure 1: Challenges in medical mathematical reasoning. (a) SFT shifts the output distribution, improving recall for seen formulas but degrading generalization. (b) Rule-based reward design struggles with continuous rewards and coarse intervals, leading to unstable precision.

domain-specific challenges (Wang et al., 2025a). On the one hand, ***supervised fine-tuning (SFT) can compromise generalization.*** By shifting the model's output distribution, SFT improves recall of training formulas but degrades performance on unseen ones, leading to knowledge forgetting (Chu et al., 2025). This phenomenon weakens robustness and limits the model's ability to generalize to new medical reasoning tasks (Figure 1 (a)). On the other hand, ***existing reward mechanisms fail to ensure stability and precision.*** Current RL-based reward designs exhibit complementary limitations: fine-grained numerical rewards, derived from continuous error signals, are noisy and difficult to optimize, often causing unstable convergence; in contrast, coarse-grained range-based rewards oversimplify evaluation, undermining precision and hindering consistent improvements in numerical accuracy (Figure 1 (b)). Furthermore, ***medical numerical reasoning demands high precision and interpretability.*** Unlike general mathematical reasoning, correctness in clinical applications cannot be verified by the final answer alone. Medical numerical reasoning requires accurate numerical predictions accompanied by transparent reasoning steps—criteria that current approaches fail to satisfy, falling short of the stringent standards of clinical safety and reliability.

To address these challenges, we propose a knowledge-guided reward framework that enhances reasoning accuracy, safety, and transparency through two complementary mechanisms. First, a **knowledge verification reward** compels models to generate explicit calculation formulas and leverages an independent verifier to check both formulas and results, mitigating knowledge degradation and enhancing interpretability during RL. Second, a **hybrid soft–hard reward** combines clinical acceptable ranges as hard constraints with progressive accuracy-based rewards as soft optimization, guiding models toward precise solutions while ensuring safety. By unifying formula verification with precision optimization, our framework offers a practical and robust approach to medical mathematical reasoning.

The contributions of this work are as follows:

- We are the first to introduce formula-level knowledge verification into the RL framework, enforcing explicit formula generation with independent validation to mitigate knowledge forgetting and significantly improve the transparency and reliability of medical mathematical reasoning.
- We propose an innovative integration of clinical safety thresholds (hard constraints) with progressive accuracy-based rewards (soft optimization), jointly optimizing safety and precision in high-risk medical scenarios.
- We conduct comprehensive experiments on MedCalc benchmark, demonstrating that our framework substantially outperforms existing methods in reasoning accuracy and knowledge robustness, while providing better generalization and clinical applicability.

## 2 RELATED WORK

**LLMs for Mathematical Reasoning.** LLMs have shown strong capabilities in complex reasoning tasks (Wang et al., 2025a; Zhang et al., 2025). Models such as GPT-4 (Achiam et al., 2023) and PaLM (Anil et al., 2023), when combined with chain-of-thought prompting (CoT) (Wei et al., 2022;

Chu et al., 2024) or tool learning (Qu et al., 2025), can effectively perform multi-step mathematical reasoning. Beyond prompting, specialized methods incorporate symbolic reasoning or external calculators to enhance arithmetic accuracy. For example, Gao et al. (2023) employs Python code generation to solve complex problems, and Imani et al. (2023) explores diverse reasoning paths to improve robustness. These approaches achieve strong results on benchmarks such as GSM8K (Cobbe et al., 2021) and MATH (Hendrycks et al., 2021). Nevertheless, most existing approaches rely primarily on final-answer verification, paying little attention to the reasoning process itself. This narrow focus limits their ability to generalize to medical numerical reasoning, where both step-wise interpretability and high precision are essential.

**RL for LLM Reasoning Tasks.** RL has emerged as a powerful approach to enhancing the reasoning capabilities of LLMs (Xu et al., 2025; Brown et al., 2024). InstructGPT (Ouyang et al., 2022) demonstrates the effectiveness of RL in aligning model behavior with human preferences. As task complexity increases, researchers explore incorporating intermediate reward signals to improve reasoning (Seed et al., 2025; Liu et al., 2025b; Jin et al., 2025; Li et al., 2025a). Lightman et al. (2024) provides supervision at intermediate reasoning steps to encourage more faithful CoT generation. Yuan et al. (2023) employs ranking-based optimization to refine output quality. In the domain of mathematical reasoning, DeepSeek-R1 (Guo et al., 2025) introduces rule-based reward mechanisms that significantly improve multi-step computation and logical consistency. Collectively, these advances suggest that RL can improve not only the quality of final answers but also the robustness and reliability of intermediate reasoning processes. Nevertheless, reward design in most RL applications remains relatively coarse. Existing methods often rely on binary rewards, scoring outputs solely based on correctness or whether they fall within an acceptable range, which fails to provide effective guidance for gradually improving numerical precision.

**Medical Mathematical Reasoning and Clinical Applications.** In the medical domain, mathematical reasoning is central to decision-making (Khandekar et al., 2024). Unlike general-purpose benchmarks (Cobbe et al., 2021; Hendrycks et al., 2021), these tasks are high-stakes, where even minor errors may compromise patient safety. Thus, clinical AI must ensure not only numerical accuracy but also transparency, interpretability, and strict safety constraints. Most prior research has focused on clinical information extraction (Ruano et al., 2025; Li et al., 2025b), medical QA (Kim & Yoon, 2025), or decision support using textual and structured data (Bahrololloomi et al., 2025; Yan et al., 2024), while largely avoiding explicit mathematical reasoning. Recent efforts to extend LLMs into clinical domains (Chen et al., 2024) show strong performance on knowledge-intensive tasks but remain limited in handling mathematical reasoning, often treating formulas as static references or generating unverified outputs. As a result, LLM-driven medical mathematical reasoning remains underexplored and insufficient for clinical reliability.

## 3 METHOD

### 3.1 PROBLEM FORMULATION

The task of medical mathematical reasoning requires extracting key clinical information from text and performing precise calculations. Formally, given an input $x = (c, q)$, where $c$ denotes the clinical context (e.g., medical records or case descriptions) and $q$ denotes the task instruction (e.g., compute creatinine clearance or adjust drug dosage), the model is expected to generate $y = (f, r, v)$, where $f$ is an explicit formula representation, $r$ denotes the step-by-step reasoning process, and $v \in \mathbb{R}$ is the numerical result. The model parameters are denoted by $\theta$, and the conditional distribution is $\pi_\theta(y \mid x)$.

Unlike general mathematical reasoning, medical mathematical reasoning requires not only accuracy relative to the ground truth $(v^*)$, but also clinical safety (i.e., results must fall within a predefined safety interval $[L, U]$) and transparency (i.e., the formula must be auditable). Thus, the overall optimization objective is:

$$\max_\theta \quad \mathbb{E}_{x \sim \mathcal{D}, \, y \sim \pi_\theta(\cdot \mid x)} \big[ R(y; x) \big] \tag{1}$$

where $R(y; x)$ combines both knowledge verification and soft–hard reward mechanisms.

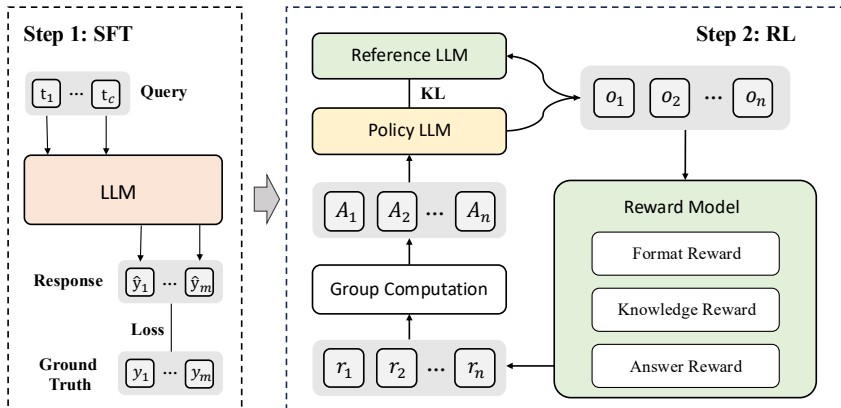

Figure 2: Overview of the proposed knowledge-guided reward framework. The framework follows a two-stage paradigm: Step 1 (SFT): the model is trained to acquire output formatting and basic medical knowledge recall; Step 2 (RL): candidate outputs are optimized using GRPO, where a reward model integrates format, formula, and answer rewards, while KL regularization ensures stable policy updates.

## 3.2 Overview of Knowledge-Guided Reward Framework

We adopt a two-stage training paradigm, as illustrated in Figure 2. SFT is first employed to learn output format and basic medical knowledge recall, followed by RL to enhance reasoning transparency and numerical accuracy. During the RL stage, we introduce a knowledge-guided reward framework consisting of three complementary components: 1. **Format Reward**: ensures that the model output follows the required structure, including explicit formula representation, step-by-step reasoning, and the final answer. Outputs that omit the formula or deviate from the required format are penalized, thereby improving auditability. 2. **Knowledge Reward**: verifies whether the generated formula aligns with domain-specific medical knowledge or task definitions, and whether its computational logic is consistent with the input conditions. This component mitigates knowledge forgetting and improves robustness. 3. **Answer Reward**: evaluates the correctness of the final numerical result. We adopt a hybrid soft–hard mechanism: a hard constraint enforces that the prediction falls within the clinically acceptable interval [L, U], while a soft reward provides progressive scores based on the deviation from the ground truth $v^*$. This combination ensures both clinical safety and precision.

$$R(y; x) = \alpha \cdot R_{\mathrm{f}}(y; x) + \beta \cdot R_{\mathrm{k}}(y; x) + \gamma \cdot R_{\mathrm{a}}(y; x) \tag{2}$$

where $\alpha, \beta, \gamma \geq 0$ balance the contributions of the three reward components.

## 3.3 Knowledge Verification Reward

During the SFT and RL stages, models often forget or generate incorrect medical formulas that do not align with task requirements. To address this issue, we introduce a formula-level knowledge verification mechanism that explicitly enforces formula recall during RL. Specifically, the model is required to generate an explicit formula $f$ as part of its output. To assess its correctness, we employ a frozen, stronger language model $\mathcal{V}$ as an external verifier. Given the input $x = (c, q)$ and the output formula $f$, the verifier determines whether $f$ belongs to the set of valid formulas $\Phi(x)$ defined for the task, and returns a binary score:

$$R_{\mathrm{knowledge}} = \begin{cases} 1.0, & \text{if } f \text{ is judged correct by } \mathcal{V}, \\ -1.0, & \text{otherwise.} \end{cases} \tag{3}$$

By explicitly enforcing formula correctness serves as a strong prior, guiding the model to recall the appropriate medical knowledge and perform calculations in a transparent and clinically reliable manner.

## 3.4 HYBRID HARD-SOFT REWARD

Medical applications impose strict safety requirements, and relying solely on coarse "acceptable range" rewards is insufficient to achieve consistent improvements in numerical accuracy. To address this, we propose a hybrid soft–hard reward mechanism that safeguards clinical safety while enhancing precision.

**Hard Constraint.** Given a predicted value $v$ and the clinically acceptable interval $[L, U]$, the hard reward is defined as:

$$R_{\text{hard}}(v; x) = \begin{cases} 2.0, & v \in [L, U], \\ -3.0, & v \notin [L, U]. \end{cases} \tag{4}$$

This binary design enforces clinical safety as a prerequisite: predictions outside the safety interval are directly penalized.

**Soft Reward.** When the ground truth $v^*$ is available, a progressive reward is further applied to encourage precision within the safety range:

$$R_{\text{soft}}(v; v^*) = \exp\left(-\frac{|v - v^*|}{\tau}\right) \tag{5}$$

where $\tau > 0$ controls sensitivity to the prediction error. The closer $v$ is to the true value, the higher the reward. The overall answer reward combines the two components additively:

$$R_{\text{answer}}(y; x) = R_{\text{hard}}(v; x) + R_{\text{soft}}(v; v^*) \tag{6}$$

This design prioritizes safety through hard constraints while simultaneously providing a smooth optimization signal for numerical precision, enabling the model to converge toward clinically reliable solutions within acceptable ranges.

## 3.5 TRAINING PROCEDURE

The overall training procedure consists of two stages: SFT and RL. Table 5 presents the training template used across experiments, and the detailed optimization procedure is summarized in Algorithm 1.

**Supervised Fine-Tuning.** In the first stage, we train the model on medical mathematical reasoning samples $(x, y^*)$ using standard teacher forcing, minimizing the cross-entropy loss:

$$\mathcal{L}_{\text{SFT}}(\theta) = -\mathbb{E}(x, y^*)\left[\sum_t \log \pi_\theta(y_t^* \mid y_{<t}^*, x)\right] \tag{7}$$

where $\pi_\theta$ denotes the policy parameterized by $\theta$. This stage ensures that the model learns the expected output format and develops a basic ability to recall relevant medical formulas.

**Reinforcement Learning.** Building upon the SFT-initialized model, we adopt a reinforcement learning framework to further improve reasoning robustness and numerical precision. Inspired by DeepSeek-R1 Guo et al. (2025), we employ Group Relative Policy Optimization (GRPO) as the optimization algorithm. GRPO improves training stability by comparing candidate outputs within the same sampling group, thereby reducing variance in gradient estimates. Concretely, for each input $x$, the model samples a set of candidate outputs $\{y_i\}_{i=1}^K$ under policy $\pi_\theta$. Each candidate is evaluated with the proposed reward function, as defined in Eq. equation 2. The policy parameters are then updated according to the GRPO gradient estimator:

$$\mathcal{J}_{\text{GRPO}}(\theta) = \mathbb{E}_{x \sim D, \{y_i\}_{i=1}^G \sim \pi_\theta}\left[\frac{1}{G}\sum_{i=1}^G \left(\min\left(P \cdot A_i, C \cdot A_i\right) - \beta\, \mathbb{D}_{\text{KL}}\left(\pi_\theta || \pi_{\text{ref}}\right)\right)\right], \tag{8}$$

$$P = \frac{\pi_\theta(y_i|x)}{\pi_{\theta_{\text{old}}}(y_i|x)}, C = \text{clip}\left(\frac{\pi_\theta(y_i|x)}{\pi_{\theta_{\text{old}}}(y_i|x)}, 1 - \epsilon, 1 + \epsilon\right), \tag{9}$$

where $A$ is the advantage value, $x$ is the input, $y_i$ is the response generated by LLMs, and $\pi_\theta(y|x) = \prod_{j=1}^{|y_i|} \pi_\theta(y_{i,j}|x, y_{i,<j})$ is the generation probability of the response $y_i$ under policy $\pi$.

## 4 EXPERIMENTS

### 4.1 DATASETS

In our experiments, we adopt the MedCalc-Bench (Khandekar et al., 2024), the first large-scale benchmark specifically designed for medical mathematical reasoning. The dataset systematically covers two major task categories: equation-based computation and rule-based reasoning, comprising a total of 55 common clinical calculation tasks. Among them, 36 are equation-based tasks and 19 are rule-based tasks. Each instance includes a clinical case description, task specification, gold-standard answer, and step-by-step computational or logical explanations, enabling comprehensive evaluation of a model's formula recall ability, clinical parameter extraction, and mathematical reasoning accuracy. Detailed statistics are presented in Table 4. The training set consists of 38 subtasks with 9,765 instances, while the test set covers 57 subtasks with 1,048 instances. Notably, the test set introduces several subtasks not present in the training set, such as additional diagnostic criteria and laboratory computations, thereby allowing evaluation of the model's cross-task generalization ability.

### 4.2 BASELINES

To comprehensively evaluate the effectiveness of our approach, we adopt a diverse set of representative LLMs as baselines. We include leading proprietary systems, GPT-3.5-Turbo and GPT-4o (Hurst et al., 2024), as upper-bound references for general reasoning, along with reasoning-oriented models such as o1-mini (Jaech et al., 2024) and DeepSeek-R1 (Guo et al., 2025), which are specifically optimized for long-chain inference. For open-source baselines, we consider a range of models including the DeepSeek-R1-Distill family (Guo et al., 2025), the Qwen2.5 series (Yang et al., 2024a), QwQ-32B (Team, 2025), and the domain-specific HuatuoGPT-o1 (Chen et al., 2024). These models span from medium to ultra-large scales and exhibit varied capabilities in mathematical reasoning and instruction following, thus serving as valuable comparative references. Finally, we include two domain-adapted variants, Qwen2.5-Instruct-1.5B and Qwen2.5-Instruct-3B, which are fine-tuned on medical mathematical reasoning tasks. These models enable us to examine the effectiveness of domain-specific adaptation in smaller-scale settings and to highlight the advantages of our proposed method over standard fine-tuning approaches.

### 4.3 IMPLEMENTATION DETAILS

In our experiments, open-source baseline models were deployed as APIs using the vLLM inference framework, with the temperature set to 1.0 and the maximum sequence length fixed at 2048; closed-source models were accessed directly via their official APIs. For SFT, we trained Qwen2.5-1.5B-Instruct and Qwen2.5-3B-Instruct with a batch size of 8 and a learning rate of $1 \times 10^{-5}$, selecting the checkpoint at 400 training steps to mitigate overfitting. During RL, we adopted GRPO as the optimization algorithm, with a batch size of 128, a learning rate of $1 \times 10^{-6}$, and a maximum response length of 2048. Each step sampled 5 rollouts, and training was conducted for 5 epochs. The formula verification reward was provided by Qwen2.5-14B-Instruct. All experiments were performed on four NVIDIA H800 GPUs with 80 GB memory each. Notably, MedCalc-R1 employs relatively small backbones, as the training data is distilled into structured key patient features rather than raw clinical narratives, reducing computational cost while enhancing efficiency and robustness in numerical reasoning. Model performance was consistently evaluated using accuracy, reflecting the proportion of predictions with correct numerical results.

### 4.4 MAIN RESULTS

Table 1 presents the main experimental results on the MedCalc-Bench dataset. Overall, closed-source models remain dominant, with DeepSeek-R1 and o1-mini representing the current upper bound of general-purpose reasoning capabilities. In contrast, existing open-source models show limited performance on medical mathematical reasoning. Even the large-scale Qwen2.5-32B-Instruct achieves only 39.03 on average, far below the closed-source counterparts. By comparison, our proposed MedCalc-R1 demonstrates significant advantages at equal or even smaller parameter scales. Specifically, MedCalc-R1$_{1.5B}$ achieves an average score of 42.36, already surpassing Qwen2.5-32B-Instruct. MedCalc-R1$_{3B}$ further improves to 51.34, obtaining the best results among open-source

| Model | Equation | | | | Rule-based | | | Avg |
|---|---|---|---|---|---|---|---|---|
| | **Lab** | **Physical** | **Date** | **Dosage** | **Risk** | **Severity** | **Diagnosis** | |
| GPT-3.5-Trubo | 21.47 | 23.24 | 8.33 | 0.00 | 21.16 | 13.75 | 25.00 | 19.85 |
| GPT-4o | 45.71 | 43.98 | 50.00 | 42.50 | 24.90 | 38.75 | 50.00 | 40.36 |
| o1-mini | 73.01 | 84.23 | 48.33 | 40.00 | 64.73 | 52.50 | 45.00 | 67.84 |
| DeepSeek-R1 | 67.48 | 93.36 | 66.67 | 57.50 | 71.78 | 56.25 | 81.67 | 73.95 |
| DeepSeek-R1-Distill-Qwen-1.5B | 1.53 | 6.64 | 1.67 | 2.50 | 2.07 | 1.25 | 1.67 | 2.86 |
| Qwen2.5-1.5B-Instruct | 1.53 | 8.10 | 1.67 | 5.00 | 5.39 | 2.50 | 12.70 | 4.82 |
| DeepSeek-R1-Distill-Qwen-7B | 5.52 | 21.26 | 11.67 | 0.00 | 10.37 | 2.50 | 16.67 | 10.78 |
| Qwen2.5-3B-Instruct | 4.29 | 13.36 | 5.00 | 0.00 | 17.84 | 16.25 | 41.27 | 12.49 |
| DeepSeek-R1-Distill-Qwen-14B | 9.82 | 26.97 | 46.67 | 5.00 | 18.26 | 13.75 | 43.33 | 19.85 |
| Qwen2.5-7B-Instruct | 19.63 | 29.55 | 18.33 | 5.00 | 19.92 | 23.75 | 47.63 | 23.37 |
| HuatuoGPT-o1♣ | 21.17 | 33.61 | 13.33 | 7.50 | 24.48 | 16.25 | 40.00 | 24.52 |
| DeepSeek-R1-Distill-Qwen-32B | 20.25 | 34.02 | 30.00 | 10.00 | 19.92 | 20.00 | 45.00 | 24.90 |
| Qwen2.5-14B-Instruct | 26.99 | 35.68 | 26.67 | 2.50 | 29.88 | 23.75 | 38.33 | 29.10 |
| QwQ-32B | 26.99 | 40.25 | 38.33 | 15.00 | 24.90 | 32.50 | 55.00 | 31.77 |
| Qwen2.5-32B-Instruct | 34.05 | 51.87 | 46.67 | 10.00 | 33.61 | 32.50 | 56.67 | 39.03 |
| Qwen2.5-1.5B-Instruct♠ | 25.46 | 54.35 | 28.33 | 32.50 | 26.97 | 10.00 | 60.00 | 33.68 |
| Qwen2.5-3B-Instruct♠ | 34.97 | 70.54 | 38.33 | 17.50 | 29.05 | 8.75 | 58.33 | 40.65 |
| MedCalc-R1$_{1.5B}$ | 36.50 | 64.73 | 26.66 | 90.00 | 26.56 | 11.24 | 73.33 | 42.36 |
| MedCalc-R1$_{3B}$ | **43.25** | **82.15** | 45.00 | 87.50 | **36.93** | 8.75 | 68.33 | **51.34** |

Table 1: Main results on the MedCalc-Bench dataset. **Bold** numbers indicate the best results, and underlined numbers denote the second-best results. ♠ marks models fine-tuned on medical numerical reasoning tasks, while ♣ refers to domain-specific medical LLMs.

models and substantially narrowing the gap with closed-source systems. These results indicate that the proposed optimization framework—combining formula verification reward and hybrid soft–hard reward—effectively enhances medical mathematical reasoning even with relatively compact models.

At the subtask level, equation-based tasks exhibit the most substantial improvements. In Dosage (drug dosage calculation), MedCalc-R1$_{1.5B}$ and MedCalc-R1$_{3B}$ achieve 90.00 and 87.50, respectively, far exceeding DeepSeek-R1 and o1-mini. This highlights the value of our reward design in high-risk clinical scenarios. Similar gains are observed in Physical and Lab, where MedCalc-R1$_{3B}$ reaches 82.15 on Physical, outperforming Qwen2.5-3B-SFT by more than 10 points. For Date-related calculations, smaller models remain unstable, but performance improves notably with larger scale. In rule-based tasks, our model achieves substantial gains in Diagnosis: MedCalc-R1$_{1.5B}$ and MedCalc-R1$_{3B}$ reach 73.33 and 68.33, respectively, approaching DeepSeek-R1. This suggests that explicit formula generation and verification enhance interpretability and reliability in clinical decision rules. However, performance on Severity remains weak, with limited improvements even after incorporating our reward mechanism. This indicates that tasks heavily dependent on discrete rule enumeration and hierarchical logic may require additional constraints on rule consistency or integration with external knowledge bases.

### 4.5 ABLATION STUDY

Table 2 reports the ablation results on the MedCalc-Bench dataset. Overall, removing either reward mechanism leads to a significant performance drop: eliminating the Knowledge Reward and the Hybrid Reward reduces the average score by 4.49% and 5.06%, respectively, while removing both results in a dramatic de-

| Model | Equation | Rule-based | Avg |
|---|---|---|---|
| MedCalc-R1 | 60.12 | 35.96 | 51.34 |
| w/o Knowledge Reward | 53.52$_{\downarrow 6.60}$ | 35.17$_{\downarrow 0.79}$ | 46.85$_{\downarrow 4.49}$ |
| w/o Hybrid Reward | 54.87$_{\downarrow 5.25}$ | 31.23$_{\downarrow 4.73}$ | 46.28$_{\downarrow 5.06}$ |
| w/o Both | 43.02$_{\downarrow 17.10}$ | 33.60$_{\downarrow 2.36}$ | 39.60$_{\downarrow 11.74}$ |

Table 2: Ablation results on MedCalc-Bench, showing the performance impact of removing the knowledge reward and hybrid reward mechanisms.

cline of 11.74%. This demonstrates that both components are necessary and complementary. Further analysis reveals that equation-based tasks are more sensitive to the knowledge reward, where

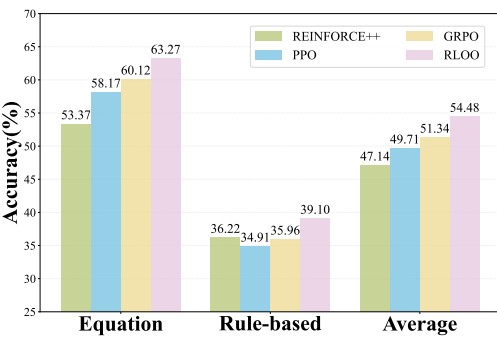
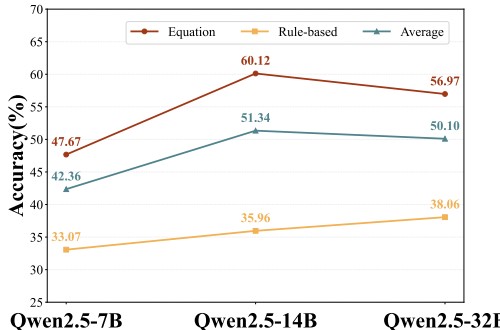

Figure 3: Performance comparison under different reinforcement learning algorithms.

Figure 4: Impact of verifier capacity for model performance.

its removal causes the largest decrease. This indicates that explicit formula generation with independent verification effectively reduces formula selection errors and knowledge forgetting, thereby enhancing stability and interpretability. In contrast, rule-based tasks benefit more from the hybrid reward, whose removal yields a larger performance drop. This confirms that the soft–hard design effectively aligns rule-based decision boundaries with clinical safety thresholds, whereas formula verification alone cannot capture such complex logic. Notably, when the Hybrid Reward is removed while retaining the Knowledge Reward, performance drops even below the configuration with both removed, further underscoring the necessity of their joint use for reliable reasoning.

### 4.6 FURTHER ANALYSIS

**Impact of Reinforcement Learning Algorithms.** The evaluation of MedCalc-R1 under four reinforcement learning optimizers (PPO , GRPO , RLOO , and REINFORCE++ ) highlights the critical role of optimization strategy in shaping both convergence and stability. As shown in Figure 3, RLOO achieves the strongest overall performance, particularly excelling in equation-based reasoning due to its variance reduction and group-relative baselines that better capture formula tokens and refine numerical precision. GRPO ranks second, consistently outperforming PPO and REINFORCE++ by leveraging group comparisons to stabilize training. PPO provides moderate but stable results, while REINFORCE++ demonstrates advantages in rule-based tasks yet suffers from instability in equation reasoning because of its aggressive exploration. These findings underscore that optimizer choice directly influences reasoning robustness and precision, with RLOO and GRPO showing the greatest alignment with the requirements of medical numerical reasoning.

**Impact of Verifier Scale.** To assess the effect of verifier capacity, we tested Qwen2.5 models of different scales (7B, 14B, 32B) as knowledge verification models. As shown in Figure 4, model size has a marked impact on performance: Qwen2.5-14B achieves the highest overall accuracy, while 32B excels on rule-based tasks but underperforms 14B on equation tasks, leading to a lower average. The 7B verifier performs worst, suggesting inadequate calibration at this scale. Task-level analysis shows equation tasks are particularly sensitive to verifier size, with accuracy improving by 12.45% from 7B to 14B but declining at 32B, likely due to overly strict judgments that induce reward sparsity. In contrast, rule-based tasks consistently benefit from larger verifiers, underscoring the value of capacity for handling complex conditions and fine-grained decision boundaries.

**Impact of Sample Size on GRPO.** To examine the effect of sample size in GRPO training, we experimented with varying rollouts (Figure 5). Performance improves notably as sample size increases, but the gain quickly diminishes: small rollouts produce high-variance gradients, while beyond five the distribution stabilizes and additional improvements are marginal relative to the cost. Task-level analysis shows that equation-based reasoning is particularly sensitive, with accuracy consistently increasing as more samples reduce reward variance and enhance numerical precision. By contrast, rule-based tasks exhibit flatter trends and larger fluctuations at small sample sizes, which are smoothed as rollouts grow. Overall, five rollouts strike the best balance between efficiency and accuracy, though larger sets may still provide incremental benefits when resources permit.

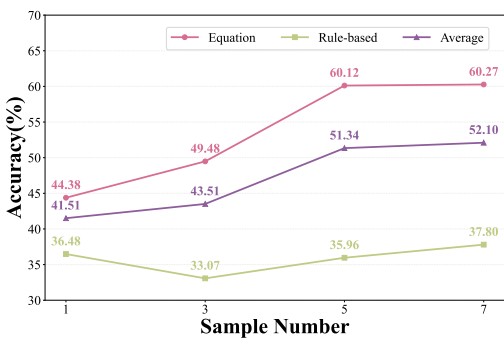

Figure 5: Effect of rollout sample size under the GRPO framework.

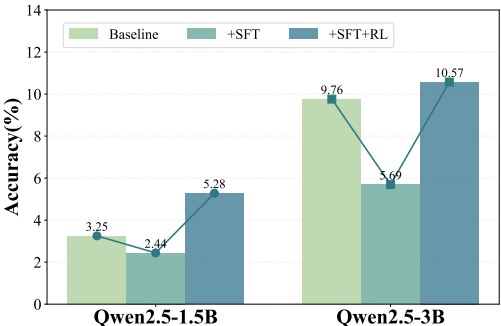

Figure 6: Cross-task generalization on unseen tasks.

### 4.7 GENERALIZATION ANALYSIS

**Generalization Across Training Methods.** To evaluate cross-task generalization, we tested different training methods on unseen tasks from MedCalc-Bench (Figure 6). Results show that combining SFT with RL substantially improves generalization across both model scales. The 3B SFT+RL model achieves the best performance, surpassing its base counterpart, while the 1.5B SFT+RL model also clearly outperforms both its base and SFT-only variants. In contrast, SFT alone weakens generalization, particularly for larger models, suggesting greater susceptibility to overfitting and distribution shift. Mechanistically, SFT tends to memorize patterns and formats but overlooks robust formula recall and precise reasoning. The RL stage introduces inductive biases independent of training distribution: formula-level verification enforces correct formula recall, reducing knowledge forgetting, while the hybrid soft–hard reward drives precise convergence under clinical safety constraints. Together, these mechanisms enhance robustness and numerical accuracy, yielding stronger transfer to unseen medical reasoning tasks.

**Generalization with Different RL Algorithms.** We further examined the impact of different reinforcement learning optimizers on model generalization (Table 3). The choice of optimizer strongly affects cross-task transferability. GRPO achieves the best performance, as its group normalization and variance reduction provide stable credit assignment that aligns well with our hybrid reward structure. REINFORCE++ benefits from stronger exploration and improves coverage on unseen cases, but its lack of a baseline leads to high gradient

| Model | Generalization |
|---|---|
| PPO | 8.54 |
| GRPO | 10.57 |
| REINFORCE++ | 10.16 |
| RLOO | 9.76 |

Table 3: Generalization results under different RL optimizers.

variance and slightly weaker performance than GRPO. RLOO performs well on in-distribution tasks but overfits local sampling distributions, limiting generalization. PPO, despite being a standard baseline, relies only on KL regularization and shows the weakest cross-task robustness.

## 5 CONCLUSION

In this work, we addressed the stringent requirements of accuracy, transparency, and safety in medical numerical reasoning. Our analysis showed that SFT often induces distributional shifts: formulas seen during training are recalled more clearly, while unseen knowledge degrades, limiting generalization. To overcome this, we proposed a knowledge-guided reward framework that integrates formula-level verification with a hybrid soft–hard reward mechanism, explicitly enhancing formula recall and reasoning transparency while jointly optimizing clinical safety and numerical precision. Experiments on the MedCalc-Bench dataset demonstrated that our framework significantly outperforms both open-source and closed-source baselines in reasoning accuracy, robustness, and cross-task generalization. Ablation and optimizer comparisons further confirmed the complementary roles of the two reward mechanisms, the suitability of GRPO, and the importance of verifier capacity in stabilizing reward signals. These findings suggest that combining structured knowledge validation with adaptive optimization is a promising direction toward developing safe, reliable, and scalable medical AI systems.

## 6 ETHICS STATEMENT

This work focuses on medical numerical reasoning using large language models. All experiments are conducted on the MedCalc-Bench dataset, which consists of synthetic and publicly available medical case descriptions without any personally identifiable information (PII). No real patient data was used, and therefore no IRB approval was required. We have carefully ensured compliance with privacy and data protection regulations, and no sensitive or private information is disclosed in this study.

We acknowledge the potential societal impact of applying LLMs in clinical domains. While our approach improves accuracy, interpretability, and safety in numerical reasoning, it is not intended for direct clinical deployment without professional oversight. Instead, the proposed framework should be viewed as a research contribution toward safer medical AI. All authors affirm adherence to the ICLR Code of Ethics throughout the research and submission process.

## 7 REPRODUCIBILITY STATEMENT

We have made extensive efforts to ensure the reproducibility of our work. The detailed description of the proposed framework, training procedure, and reward design can be found in Section 3. Complete implementation details, including hyperparameter settings, training configurations, and evaluation protocols, are provided in Section 4 and the Appendix. For reproducibility, we will release the code, training scripts, and instructions for dataset preprocessing as anonymous supplementary material. All proofs, assumptions, and derivations of the theoretical formulations are included in the appendix. Together, these materials enable independent researchers to replicate our experiments and verify our results.

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

| Task Type | Subtask Type | Train Dataset | | Test Dataset | |
|---|---|---|---|---|---|
| | | #Task | #Inst. | #Task | #Inst. |
| **Equation-based** | Lab | 16 | 2,931 | 19 | 326 |
| | Physical | 12 | 4,804 | 13 | 241 |
| | Date | 3 | 240 | 3 | 60 |
| | Dosage | 2 | 151 | 2 | 40 |
| **Rule-based** | Risk | 4 | 1,206 | 13 | 241 |
| | Severity | 1 | 75 | 4 | 80 |
| | Diagnosis | 3 | 358 | 3 | 60 |
| **Overall** | | 38 | 9,765 | 57 | 1,048 |

Table 4: Dataset statistics of MedCalc-Bench. The benchmark covers diverse equation-based and rule-based tasks.

---

You are a helpful assistant. The user asks a medical calculation question, and the Assistant solves it. The assistant first recalls the required formulas or scoring standards and thinks about the reasoning process in the mind, and then provides the user with the answer. The formulas, reasoning process, and final answer are enclosed within `<formula> </formula>`, `<think> </think>` and `<answer> </answer>` tags, respectively, i.e., `<formula>` medical formulas or scoring standards `</formula>` `<think>` reasoning process here `</think>` `<answer>` answer here `</answer>`. Now, the user will provide you with key information about a patient and ask you to solve a calculation reasoning problem based on that information. After thinking, when you finally arrive at an answer, place the result within `<answer> </answer>` tags. Here is the patient key note: {patient_key_note}. Here is the question: {question}. Let me solve this step by step.

---

Table 5: Template for question and patient_key_note will be replaced with the specific question and patient note during training and inference.

## A  LLM Usage Statement

We used LLMs only as auxiliary tools for translation, grammar polishing, and minor wording improvements. No LLMs were involved in research ideation, experiment design, data analysis, or result interpretation. All methodological development, experiments, and scientific contributions were conducted solely by the authors.

## B  Case Study

To further validate the reasoning ability of the models, we conducted in-depth case analyses, with representative examples shown in Figures 7 and 8. The results reveal that Qwen2.5-3B-Instruct exhibits clear deficiencies in medical numerical reasoning. Specifically, its formula recall is unstable, often leading to omissions or incorrect invocation of standard clinical equations. In addition, the generated reasoning chains are relatively short and lack step-by-step calculations, which undermines transparency and auditability. Such limitations are particularly concerning in high-risk clinical tasks, where the absence of explicit intermediate steps makes errors more difficult to identify and correct.

In contrast, our proposed MedCalc-R1, after incorporating the knowledge-guided reward framework, demonstrates substantial advantages. (1) In terms of formula recall, the model reliably generates clinically valid formulas while avoiding common omissions and misapplications. (2) Regarding the reasoning process, MedCalc-R1 produces more complete chains, encompassing explicit formula substitution, progressive calculations, and the final result, thereby enhancing transparency and interpretability. (3) For numerical accuracy, the model not only produces correct final answers but also ensures consistency across intermediate steps, significantly reducing potential computational errors.

---

**Creatinine Clearance (Cockcroft-Gault Equation)**

**Patient Note:**

In 2008, a 59-year-old Japanese woman was admitted for evaluation of renal disease. RA had been diagnosed at another hospital in 1972 when she presented with bilateral arthropathy of the hands, knees, ankles, and feet. Treatment was started with a combination of a gold preparation and nonsteroidal anti-inflammatory drugs (NSAIDs), but was not been effective. Prednisolone (PSL; 15 mg daily) and bucillamine (BUC; 200 mg daily) were started in 1987, but her disease remained active. Methotrexate (MTX; 5 mg daily) was started in 1995 but was discontinued because of nausea. In 2002, urinary protein was found to be positive by a dipstick urine test, and BUC was stopped. Then treatment was continued with PSL (5 mg/day) and loxoprofen (50 mg/day). However, urinary protein excretion increased in 2007, and serum creatinine (Cre) was elevated to 1.96 mg/dL.\nOn admission, the patient was 154.2 cm tall and weighed 44.0 kg, with a blood pressure of 128/60 mmHg and temperature of 36.4 °C. Physical examination did not reveal any abnormalities of the heart and lungs. The joints of her hands, knees, ankles, and feet showed bilateral swelling and deformity. In addition, the lower extremities were edematous. Her cervical spine was unstable, with flexion causing numbness in the upper limbs.\nLaboratory findings were as follows: serum Cre was 4.2 mg/dL, the estimated glomerular filtration rate (eGFR) was 9.3 mL/min/1.73m3, C-reactive protein (CRP) was 0.9 mg/dL, and SAA was 43.2. In addition, rheumatoid factor (RF) was positive at 59 U/mL (normal: < 10), and cyclic citrullinated peptide (CCP) antibodies were positive at 218.5 (normal < 4.5). 24-hour urinary protein excretion was 6.5 g, and the urine sediment contained 1 – 5 red cells per high-power field (HPF). The disease activity score (DAS)-CRP was 7.1. Radiographs showed deformation of the finger and foot joints as well as atlantoaxial joint subluxation. Renal biopsy was performed for evaluation of her kidney disease.\nRenal biopsy\nLight microscopic examination of a biopsy specimen containing 4 glomeruli revealed global sclerosis in all 4. There was severe tubular atrophy, and tubulointerstitial fibrosis occupied ~ 95% of the entire renal cortex. All 4 glomeruli contained multinodular structures of amorphous material with a PAM-positive border. This material was positive for Congo-red and amyloid A, but was negative for κ and λ chains, β-2 microglobulin, and transthyretin (). Electron microscopy showed randomly arranged fibrils measuring 8 – 12 nm in diameter corresponding to the amyloid deposits (f). AA amyloidosis was diagnosed from these findings. In addition to the glomeruli, amyloid deposits were mainly observed in the interlobular artery walls and tubulointerstitium (e). Endoscopic biopsy of the stomach, duodenum, and colon revealed AA-positive deposits in the small arteries and tissues of the submucosal layer (a).\nClinical course\nPSL was discontinued, and administration of a soluble tumor necrosis factor (TNF) receptor inhibitor (etanercept; 25 mg every 2 weeks) was started in May 2008, but it was not effective. By September 2008, Cre was increased to 6.0 mg/dL. She underwent surgery to prepare an arteriovenous fistula for hemodialysis. Etanercept was discontinued, and a humanized anti-interleukin-6 receptor antibody (tocilizumab; 8 mg/kg = 360 mg/month) was started in February 2009. After 3 months, her CRP decreased to 0.0 mg/dL, and the DAS28-CRP sore was 2.12. After 2 years of tocilizumab therapy, urinary protein excretion was decreased to 1.1 g/day, and Cre was 4.0 mg/dL. Subsequently, Cre remained in the range of 4.5 – 5.0 mg/dL until December 2017. While Cre increased to 7.1 mg/dL after initiation of treatment with denosumab (a human monoclonal antibody that binds to receptor activator of NFκB ligand) for osteoporosis in October 2018, it remained at 7.0 mg/dL in June 2019 ().\nGastroduodenal biopsy was performed in May 2013 and May 2017. On both occasions, no amyloid deposits were detected in the submucosal blood vessels (b).

**Relevant Entities:**

{'sex': 'Female', 'age': [59, 'years'], 'weight': [44.0, 'kg'], 'height': [154.2, 'cm'], 'creatinine': [4.2, 'mg/dL']}

**Question:**

What is the patient's Creatinine Clearance using the Cockroft-Gault Equation in terms of mL/min? You should use the patient's adjusted body weight in kg instead of the patient's actual body weight if the patient is overweight or obese based on their BMI. If the patient's BMI's normal, set their adjusted body weight to the minimum of the ideal body and actual weight. If the patient is underweight, please set their adjusted body weight to their actual body weight. You should use the patient's medical values and health status when they were first admitted to the hospital prior to any treatment.

**Ground Truth Answer:** 10.01786

**Lower Limit:** 9.51697

**Upper Limit:** 10.51875

**Qwen2.5-3B-Instruct:**

First, calculate the BMI: weight / (height * height) = 44 / (1.54 * 1.54) = 18.1. Since the BMI is less than 25, it is normal. So the adjusted body weight should be the minimum of the ideal body weight and the actual body weight. Next, use the ideal body weight of 60kg and the creatinine value of 4.2 to calculate the creatinine clearance using the Cockcroft-Gault equation: **(140 - age) * weight in Hl * 0.85 / [72 * serum creatinine in mg/dL]** = (140 - 59) * 60 * 0.85 / [72 * 4.2] = 12.76. So the patient's creatinine clearance is 12.76\", \"answer\": \"**12.76**\".

**MedCalc-R1 Response:**

**</formula>**The formula for computing Cockcroft-Gault is given by CrCl = ((140 - age) * adjusted weight * (gender_coefficient)) / (serum creatinine * 72), where the gender_coefficient is 1 if male, and 0.85 if female. The serum creatinine concentration is in mg/dL.**</formula><think>**\nThe patient's gender is female, which means that the gender coefficient is 0.85.\nThe patient is 59 years old. \nThe concentration of creatinine is 4.2 mg/dL. \nThe formula for computing the patient's BMI is (weight)/(height * height), where weight is the patient's weight in kg and height is the patient's height in m.\nThe patient's height is 154.2 cm, which is 154.2 cm * 1 m / 100 cm = 1.542 m. The patient's weight is 44.0 kg. The patient's bmi is therefore 44.0 kg / (1.542 m * 1.542 m) = 18.718 kg/m^2.The patient's BMI is 18.7, indicating they are normal weight.\nBecause the patient is normal, we take take minimum of the ideal body weight and the patient's body as the patient's adjusted weight for the Cockcroft-Gault Equation. Hence, the adjusted body weight is the minimum of the two giving us an adjusted body weight of 44.0 kg.\n\nUsing the Cockcroft-Gault equation:\nCrCl = ((140 - age) * adjusted weight * gender_coefficient) / (serum creatinine * 72).\nPlugging the patient's values gives us ((140 - 59) * 44.0 * 0.85) / (4.2 * 72) = 10. susceptible ml/min. Hence, the patient's creatinine clearance is 10.0 mL/min.**</think><answer>**10**</answer>**

Figure 7: Case study on creatinine clearance calculation using the Cockcroft–Gault equation.

---

**CHA2DS2-VASc Score for Atrial Fibrillation Stroke Risk**

**Patient Note:**

An 86-year-old woman with a history of atrial fibrillation, aortic stenosis, and hypertension presented to the emergency department due to altered mental status. She was lethargic, confused, and was not answering questions appropriately for the past four days. On arrival, she was afebrile, and her vitals included a pulse of 139 beats per minute, blood pressure of 110/79 mmHg, and a respiratory rate of 16 breaths per minute with saturation of 96% on room air. Physical exam was significant for a slow to respond female, orientated to self, who was able to follow commands with no focal neurological deficit. Her skin was warm and well-perfused with normal capillary refill, with no rashes or petechiae. The cardiovascular exam was significant for tachycardia, an irregular heart rhythm, and a systolic murmur heard best at the right upper sternal border. She had lower extremity pitting edema bilaterally. Her lab results were notable for elevated white blood cell count (WBC) 25.2 x 103 μL (3.9-11.3 x 103 μL), hemoglobin 11.0 g/dL (11.3-15.1 g/dL), platelets 58.0 x 103 μL (165-366 x 103 μL), troponin 0.164 ng/mL (<0.010 ng/mL), lactic acid 2.5 mmol/L (0.5-2.0 mmol/L), and a basic chemistry panel was within normal limits. The urinalysis was remarkable for the presence of WBCs, leukocyte esterase, and bacteria with a urine culture pending. Additionally, two sets of blood cultures were obtained. Her electrocardiogram was significant for atrial fibrillation with a rapid ventricular rate (RVR). She was started on ceftriaxone for probable UTI and was admitted to the hospital for further management. Further diagnostics included lower extremity Doppler ultrasound, which revealed deep venous thrombosis (DVT). A bedside echocardiogram indicated right ventricular strain, and the N-terminal proB-type natriuretic peptide level was elevated at 8,215 pg/mL (<449 pg/mL). Due to concern for possible pulmonary embolism in the setting of known DVT, the patient was started on a continuous heparin infusion. A transthoracic echocardiogram revealed severe aortic valve calcification with new aortic insufficiency. Blood and urine cultures returned positive for Aerococcus urinae. A transesophageal echocardiogram further confirmed moderate to severe aortic regurgitation with findings suggestive of infective endocarditis (IE), see Figures , .\nAntibiotics were expanded to include gentamicin with ceftriaxone for IE, and surgical evaluation was consulted. Given she was a high-risk surgical candidate, surgery intervention was not offered at this time unless evidence of further clinical decompensation. Additional surgical recommendations included re-evaluation after completion of antibiotic course. She was therefore discharged on intravenous antibiotics to complete a total of six weeks of treatment with ceftriaxone and two weeks of gentamicin. Soon after completion, the patient returned due to worsening thrombocytopenia that had previously improved upon discharge. On readmission, repeat labs were most notable for worsening thrombocytopenia, leukocytosis, acute kidney injury, and hyperkalemia. The patient was in atrial fibrillation with RVR and physical exam concerning for decompensated heart failure. Despite administration of broad-spectrum antibiotics and heart rate control, the patient continued to decline, and her clinical picture was consistent with cardiogenic shock. In addition to respiratory distress, the patient went into ventricular fibrillation arrest. Due to the patient's wishes to not be intubated or resuscitated, she was made comfortable and passed away.

**Relevant Entities:**

{'sex': 'Female', 'age': [86, 'years'], 'Congestive Heart Failure': True, 'Thromboembolism history': False, 'Vascular disease history': False, 'Hypertension history': True}

**Question:**

What is the patient's CHA2DS2-VASc Score? You should use the patient's medical values and health status when they were first admitted to the hospital prior to any treatment.

**Ground Truth Answer:** 5

**Lower Limit:** 5

**Upper Limit:** 5

**Qwen2.5-3B-Instruct:**

First, I check the key question which is the patient's CHA2DS2-VASc Score. The formula is as follows: **2 x sex (Score of 1 if Female, Score of 0 if Male) + age divided by 20 if over 75, 1 if 70-74, 0 if 50-69, -1 if 45-59, 0 if 40-44.** In this case, the patient is a female, and she is 86 years old.\", \"answer\": \"**4**\"

**MedCalc-R1 Response:**

**</formula>**The criteria for the CHA2DS2-VASc score are listed below:\n\n1. Age: < 65 years = 0 points, 65-74 years = +1 point, ≥ 75 years = +2 points\n2. Sex: Female = +1 point, Male = 0 points\n3. Congestive Heart Failure (CHF) history: No = 0 points, Yes = +1 point\n4. Hypertension history: No = 0 points, Yes = +1 point\n5. Stroke, Transient Ischemic Attack (TIA), or Thromboembolism history: No = 0 points, Yes = +2 points\n6. Vascular disease history (previous myocardial infarction, peripheral artery disease, or aortic plaque): No = 0 points, Yes = +1 point\n7. Diabetes history: No = 0 points, Yes = +1 point\n\nThe CHA2DS2-VASc score is calculated by summing the points for each criterion. **</formula><think>**\n \nThe current CHA2DS2-VASc score is 0.\nThe patient is 86 years old. Because the age is greater than 74, two points added to the score, making the current total 0 + 2 = 2.\nThe patient's gender is female and so one point is added to the score, making the current total 2 + 1 = 3.\nThe patient history for congestive heart failure is present. Because the patient has congestive heart failure, one point is added to the score, making the current total 3 + 1 = 4.\nThe patient history for hypertension is present. Because the patient has hypertension, one point is added to the score, making the current total 4 + 1 = 5.\nOne criteria of the CHA2DS2-VASc score is to check if the patient has had any history of stroke, transient ischemic attacks (TIA), or thromboembolism. Because stroke history is not specified in the patient note, we assume that it is absent from the patient. Because tia history is not specified in the patient note, we assume that it is absent from the patient. Based on the patient note, the patient history for thromboembolism is absent. Because all of stroke, tia, or thromboembolism are absent, no points are added to score, keeping the score at 5.\nBased on the patient note, the patient history for vascular disease is absent. Because the patient does not have vascular disease, no points are added to score, keeping the score at 5. Because diabetes history is not specified in the patient note, we assume it's value as 'absent'. Because the patient does not have diabetes, no points are added to score, keeping the score at 5.\nThe patient's CHA2DS2-VASc Score is 5. **</think><answer>**5**</answer>**

Figure 8: Case study on atrial fibrillation stroke risk assessment using the CHA2DS2-VASc score.

---

**Algorithm 1** Knowledge-Guided Training with GRPO

---

**Require:** Dataset $\mathcal{D} = \{(x, v^*, [L, U])\}$ with input $x = (c, q)$, ground-truth value $v^*$ and clinical safety interval $[L, U]$; policy $\pi_\theta$; reference policy $\pi_{\text{ref}}$; weights $\alpha, \beta, \gamma$; temperature $\tau$; group size $K$; GRPO clip $\epsilon$; KL weight $\lambda_{\text{KL}}$

**Ensure:** Trained policy $\pi_{\theta'}$

1: **Stage I: Supervised Fine-Tuning (SFT)**
2: **for** epoch $= 1, \ldots, E_{\text{SFT}}$ **do**
3:     **for** mini-batch $\mathcal{B} \subset \mathcal{D}$ **do**
4:         Update $\theta \leftarrow \theta - \eta \nabla_\theta \big[ -\sum_{(x,y^*) \in \mathcal{B}} \sum_t \log \pi_\theta(y_t^* \mid y_{<t}^*, x) \big]$
5:     **end for**
6: **end for**

7: **Stage II: RL with Knowledge-Guided Reward (GRPO)**
8: **for** epoch $= 1, \ldots, E_{\text{RL}}$ **do**
9:     **for** mini-batch $\mathcal{B} \subset \mathcal{D}$ **do**
10:         **for** each $x \in \mathcal{B}$ **do**
11:             Sample a group of $K$ candidates $\{y_i = (f_i, v_i)\}_{i=1}^K \sim \pi_\theta(\cdot \mid x)$
12:             **for** $i = 1$ to $K$ **do**
13:                 $R_{\text{format}} \leftarrow \text{FORMATREWARD}(y_i)$
14:                 $R_{\text{knowledge}} \leftarrow \text{KNOWLEDGEREWARD}(f_i, x)$
15:                 $R_{\text{answer}} \leftarrow \text{ANSWERREWARD}(v_i, v^*, [L, U], \tau)$
16:                 $R_i \leftarrow \alpha R_{\text{format}} + \beta R_{\text{knowledge}} + \gamma R_{\text{answer}}$
17:             **end for**
18:             $\bar{R} \leftarrow \frac{1}{K} \sum_{i=1}^K R_i$                                            ▷ group baseline
19:             **for** $i = 1$ to $K$ **do**
20:                 $A_i \leftarrow R_i - \bar{R}$                            ▷ group-relative advantage
21:                 $r_i \leftarrow \frac{\pi_\theta(y_i|x)}{\pi_{\theta_{\text{old}}}(y_i|x)}$                      ▷ importance ratio
22:                 $r_i^{\text{clip}} \leftarrow \text{clip}(r_i, 1 - \epsilon, 1 + \epsilon)$
23:                 $\mathcal{L}_i \leftarrow -\min\{r_i A_i, \ r_i^{\text{clip}} A_i\} + \lambda_{\text{KL}} \text{KL}\big(\pi_\theta(\cdot \mid x) \,\|\, \pi_{\text{ref}}(\cdot \mid x)\big)$
24:             **end for**
25:             Update $\theta \leftarrow \theta - \eta \nabla_\theta \frac{1}{K} \sum_{i=1}^K \mathcal{L}_i$
26:         **end for**
27:     **end for**
28: **end for**
29: **return** $\pi_{\theta'}$

---

