# OpenReview forum: "MedCalc-R1: Knowledge-Guided Reward Framework for Medical Mathematical Reasoning"
_ICLR.cc/2026/Conference — ICLR 2026 Conference Withdrawn Submission_

### Official Review · Reviewer_dW9S · 2025-10-27

**Soundness:** 2
**Presentation:** 2
**Contribution:** 2
**Rating:** 2
**Confidence:** 4

**Summary:**

The authors present a reinforcement learning framework tailored for medical mathematical reasoning, which integrates a knowledge verification reward and a hybrid answer reward. Experiments conducted on the MedCalc benchmark demonstrate that the proposed method outperforms existing baselines. Further analyses are provided to investigate the contributions of individual components.

**Strengths:**

- The paper proposes a domain-specific reinforcement learning framework for medical mathematical reasoning, which jointly considers both the validity of formulas and the correctness of final answers in the reward design.
- Experimental results on MedCalc-Bench show that the proposed method achieves superior performance compared to several baselines.

**Weaknesses:**

- The authors emphasize that medical mathematical reasoning requires high interpretability through transparent reasoning steps, and claim that the proposed knowledge verification mechanism significantly enhances transparency and reliability. However, there is no explicit evaluation or quantitative analysis of interpretability in the experiments, which makes this claim less substantiated.
- The proposed approach depends on clinically acceptable intervals to compute the hard reward. However, it remains unclear how these intervals are determined. Although MedCalc-Bench provides lower and upper bounds for each question, they are set to 95%–105% of the ground-truth value for equation-based calculators and identical to the ground truth otherwise. Such a setting may not always ensure clinical safety. Therefore, it would be helpful for the authors to clarify how these intervals are obtained—whether they are manually labeled (and if so, at what cost) or automatically generated (and what strategies or validation methods are used). The sensitivity of the model’s performance to potential inaccuracies in these intervals should also be discussed.
- There are several issues in Table 1. While the caption states that bold font denotes the best performance and underlining indicates the second-best, only the last row actually follows this convention—many other rows lack proper highlighting. Moreover, some bolded values do not correspond to the highest scores; for instance, in the “Avg” column, the R1 model achieves the best result, yet the proposed method is incorrectly highlighted. This might lead to misunderstanding regarding the relative performance of the proposed approach.
- There are two experiments (Figure 3 and Table 3) comparing different reinforcement learning strategies. However, since this work does not propose a new reinforcement learning algorithm but rather a reward mechanism tailored for medical mathematical reasoning, it is unclear how these experiments demonstrate the advantages of the proposed method.

**Questions:**

- The computation of the format reward is not clearly described and would benefit from additional clarification.
- It would be valuable to discuss the computational cost of the proposed method, as the knowledge verifier model appears to be substantially larger, potentially increasing the overall computational burden. Additionally, if the method were applied to a larger backbone model, would it require an even larger verifier model? If so, this could raise concerns about scalability.
- The settings of several hyperparameters are not specified. The values of $\alpha$, $\beta$, $\gamma$, and $\tau$ are not provided. Moreover, while the paper discusses the influence of sample size in GRPO, it remains unclear how other hyperparameters (e.g., learning rate) are selected—whether through grid search or empirical tuning. The absence of such details may affect reproducibility.
- Figure 3 shows that RLOO substantially outperforms GRPO, yet the main results are reported using GRPO. It would be helpful to clarify the rationale for this choice.
- In the conclusion, the authors state that their method significantly outperforms both open-source and closed-source baselines in reasoning accuracy. However, according to Table 1, the proposed approach performs worse than DeepSeek-R1 and o1-mini on average accuracy, which appears inconsistent with that claim.

---

### Official Review · Reviewer_KHHK · 2025-10-28

**Soundness:** 2
**Presentation:** 2
**Contribution:** 2
**Rating:** 2
**Confidence:** 5

**Summary:**

The paper introduces MEDCALC-R1, a framework designed to improve the accuracy, safety, and transparency of Large Language Models (LLMs) in high-stakes medical mathematical reasoning. It addresses existing model failures, such as "knowledge degradation" and computational bias, by implementing a novel two-part reinforcement learning (RL) reward system. This system includes a knowledge verification reward, which forces the model to explicitly generate a formula that is then checked by an independent verifier , and a hybrid soft-hard reward, which heavily penalizes answers outside a clinical safety range (hard constraint) while also rewarding answers for their precise proximity to the correct value (soft optimization).

**Strengths:**

The framework's primary strength is its reward design, which is the first to integrate explicit formula-level verification directly into an RL process and uniquely combines strict clinical safety constraints with fine-grained numerical precision. This approach leads to strong empirical performance, enabling smaller MedCalc-R1 models to significantly outperform much larger open-source baselines, especially in critical tasks like drug dosage calculation.

**Weaknesses:**

The "Soft Reward" design is counter-intuitive and appears unjustifiable. Specifically, for the task of calculating medical metrics from a given patient's Electronic Health Record (EHR), there should be a definitive standard answer. The LLM's output is binary—either correct or incorrect. In such a scenario, the introduction of a Soft Reward mechanism seems unnecessary and lacks a strong rationale. Notably, the paper references DeepSeek R1, whose most significant contribution was achieving SOTA performance on analogous calculation tasks (which have ground truths) by employing a Rule-Based Hard Reward. This paper's approach seems to be in direct opposition to the core findings of DeepSeek R1.

It is unclear why the authors employed a surprisingly weak LLM (Qwen 2.5 32B) as the External Verifier. As illustrated in Figure 4, this verifier appears largely incapable of accurately assessing the correctness of the generated formulas. The authors must justify why they opted against using demonstrably stronger LLMs, such as DeepSeek R1, GPT-o1, Qwen3-235B-A22B, or Qwen2.5-72B, which would presumably yield better verification performance.

The paper defines an "acceptable range" for predicted values (L and U), yet it seemingly fails to explain how these bounds are determined in practice. Furthermore, in Subsection 3.3, the authors use the expression "belongs to the set of valid formulas Φ(x) defined for the task." My understanding is that Φ(x) should represent a single, unique solution. The calculation of a specific medical metric or risk score typically relies on one established formula, not a "set of functions" as the notation implies.

The proposed model's performance is significantly inferior to DeepSeek R1. While this is likely because the model trained in this study is smaller, the fact remains that hospitals can deploy DeepSeek R1 locally to achieve superior results. This reality substantially undermines the practical significance and contribution of this work.

Following on from the first point (regarding the reward design), the paper's contribution appears to be limited to a simple modification of the Reward Function applied to an off-the-shelf RL algorithm. Although this modified reward function shows some utility, the paper lacks a more granular or deeper analysis of its effects.

**Questions:**

Why the authors adopted a weak LLM (Qwen 2.5 32B) as the External Verifier?

---

### Official Review · Reviewer_zhwt · 2025-10-31

**Soundness:** 3
**Presentation:** 3
**Contribution:** 2
**Rating:** 4
**Confidence:** 3

**Summary:**

This paper focuses on medical calculation tasks (e.g., drug dosage, clinical score, and physiological index estimation) and proposes a knowledge-guided training and evaluation framework that integrates formula-based verification and reward design. The method is evaluated only on a single dataset, showing competitive results for small-scale models (1.5B and 3B). Overall, the paper is clearly written and easy to follow.

**Strengths:**

S1  The target task is of high clinical importance since computational accuracy directly affects treatment safety and risk assessment.

S2  The methodology and experiments are presented in a logically coherent manner and can be followed without major difficulty.

S3  The paper emphasizes explicit formula-based reasoning and verifiable outputs, aligning well with the need for traceability.

**Weaknesses:**

W1 The authors use the term Medical Mathematical Reasoning, which is rarely seen in recent literature. The task described essentially overlaps with Medical/Clinical Calculation, a term widely adopted by prior works and benchmarks. If the task is equivalent, adopting the mainstream naming would make the work more discoverable and consistent with community standards.

W2 The paper claims improvements for small models (1.5B/3B) while acknowledging that larger general-purpose models still perform better overall, which is expected. However, the comparison set is limited: It lacks specialized small-scale models that are domain-specific or distilled for medical computation. It therefore cannot demonstrate whether the reported advantage truly arises from the proposed framework or simply from scale limitations. A fair evaluation should include both larger foundation models and smaller, domain-specific models trained under comparable conditions.

W3  The entire study relies on one dataset, without cross-validation on other benchmarks such as CalcQA[1], which represent different task forms. Using only one dataset raises concerns about overfitting and limits the claim of general applicability.

W4 The paper reports only one evaluation metric, which is not rigorous for medical contexts. Optimizing a single objective may bias the model toward that metric while neglecting clinically relevant dimensions. It is crucial to include additional metrics to better reflect real-world reliability.

W5 The paper does not provide details on the prompt templates or sampling parameters. Prompt design significantly affects numerical reasoning, as highlighted in [2], which showed large performance gaps between well-designed and poorly designed prompts. Lack of prompt transparency weakens reproducibility and interpretability.

W6 Recent studies on medical calculation, verifiable rewards, process supervision, and benchmarks are not sufficiently discussed, leaving the positioning of this paper unclear.

W7 The paper omits convergence curves, variance analyses, or computational cost of reward verification, making it difficult to assess training stability or real deployment feasibility.

[1]Zhu, Yakun, et al. "MeNTi: Bridging Medical Calculator and LLM Agent with Nested Tool Calling." Proceedings of the 2025 Conference of the Nations of the Americas Chapter of the Association for Computational Linguistics: Human Language Technologies (Volume 1: Long Papers). 2025.

[2]Goodell, Alex J., et al. "Large language model agents can use tools to perform clinical calculations." npj Digital Medicine 8.1 (2025): 163.

**Questions:**

Please see weaknesses.

---

### Official Review · Reviewer_ezs7 · 2025-11-01

**Soundness:** 3
**Presentation:** 3
**Contribution:** 2
**Rating:** 4
**Confidence:** 4

**Summary:**

The paper proposes MedCalc‑R1, a two‑stage post‑training framework for medical mathematical reasoning. After supervised fine‑tuning (SFT) to learn output format and recall, the method applies reinforcement learning with a knowledge‑guided reward that has three parts: (i) a format reward enforcing outputs that include an explicit formula, step‑wise reasoning, and final value; (ii) a knowledge verification reward that compels the model to generate a valid clinical formula and uses a stronger frozen LLM as an external verifier; and (iii) a hybrid answer reward that combines a hard clinical safety interval with a soft, accuracy‑graded term. The RL stage uses GRPO with KL regularization. On MedCalc‑Bench, a 55‑task benchmark spanning equation‑based and rule‑based clinical calculations , MedCalc‑R1 (1.5B/3B backbones) outperforms open‑source baselines and narrows the gap to proprietary systems; e.g., the 3B model achieves 51.34 average accuracy vs 39.03 for Qwen2.5‑32B‑Instruct. Ablations show both the knowledge verifier and the hybrid reward are necessary, and further analyses examine optimizer choice (RLOO/GRPO), verifier scale, and rollout count. Case studies illustrate transparent, auditable outputs.

**Strengths:**

1. Well‑motivated objective: medical calculations require precision, transparency, and safety. The design directly encodes these needs.
2. Method simplicity and practicality: integrates cleanly with standard RL for LLMs (GRPO) and does not require complex tooling; verifier is plug‑replaceable.
3. Substantive empirical gains on MedCalc‑Bench, including outperforming much larger open‑source models; strong results on Dosage and Physical subtasks.
4. Careful analyses: ablations for both mechanisms, optimizer comparison, verifier capacity study, rollout sensitivity, and cross‑task generalization.
5. Transparency: case studies show explicit formulas and computations, aligning with the paper’s interpretability goal.

**Weaknesses:**

1. Verifier specification and reliability. The formula‑verification reward hinges on an LLM judge. The paper does not quantify verifier accuracy vs. ground‑truth formula annotations, inter‑rater reliability, or the impact of verifier mistakes on training stability. It is also unclear how (\Phi(x)) is defined and enforced across 55 tasks (e.g., synonymous formulas, unit variants, edge cases). Providing a measured verifier error rate and robustness to noisy judgments would strengthen claims.

2. Format reward under‑specified. "FORMATREWARD" is referenced but not defined with an exact scoring schema (regex/spec parsing, weighting, penalties). Reproducibility would benefit from a formal grammar and ablations on this component alone.

3. Safety interval dependence. The hard reward uses per‑item ([L,U]). Although this fits the benchmark (case card shows lower/upper limits), practical deployment may not supply such intervals, and they may be guideline‑dependent. A sensitivity study varying ([L,U]) or learning them from task definitions would clarify generality.

4. Interpretability not quantitatively evaluated. While format and verification aim to enhance transparency, the paper does not include process‑level metrics (e.g., formula correctness rate, step‑consistency checks, auditor agreement). The case studies are illustrative but anecdotal.

5. Limited text‑to‑structure realism. Inputs are structured key patient features rather than raw notes; this reduces complexity and may overstate clinical readiness. An end‑to‑end pipeline with clinical text extraction would be informative.

6. Statistical reporting. Main tables report point accuracies without confidence intervals or multi‑seed variance; some gains are large, but dispersion would help assess stability.

**Questions:**

1. Verifier details: How is (\Phi(x)) (the set of valid formulas) constructed for each subtask? Are symbolic templates or unit‑normalized forms used? What is the observed false‑positive/false‑negative rate of the LLM verifier relative to human annotations across a held‑out set?

2. Format reward rubric: Please provide the exact scoring (parsing rules, penalties for missing sections, weighting) for "FORMATREWARD". Could you add an ablation "w/o format reward" to disentangle its effect from the other two components?

3. Clinical intervals: Are ([L,U]) always available in MedCalc‑Bench, and how are they derived? How sensitive are results to widening/narrowing these intervals, or replacing them with published clinical thresholds when multiple guideline variants exist?

4. Verifier robustness: What happens if the verifier is smaller/weaker or adversarially perturbed (e.g., small wording changes)? Can you stabilize training with soft verification scores (confidence/logits) instead of a hard ({+1,-1}) signal?

5. Process metrics: Can you report formula‑correctness rate and step‑consistency metrics (e.g., recomputing the final answer from the emitted steps) to substantiate interpretability gains beyond final accuracy?

6. Variance and significance: Please include multi‑seed means/standard deviations (or CIs) for Table 1 and Table 2, and clarify the number of runs used for the optimizer comparisons in Fig. 3.

7. Scalability: What are the compute and latency overheads of verifier calls during RL and at inference (if any)? Could self‑verification or distilled verifiers reduce cost without hurting accuracy?

8. Distribution shift: Since inputs are structured, do results hold if you prepend short natural‑language case narratives or vary units (mg/dL vs µmol/L)? Any experiments on unit normalization errors?

9. Would you please benchmark state-of-the-art reasoning LLMs, e.g., GPT-5-high, Gemini-2.5-pro, Claude 4.5-thinking, etc.? If their performance is already excellent, that might indicate the problem is largely solved, especially considering that the compute used here is negligible compared with the massive post-training compute invested in those SOTA reasoning models.

---

### Official Review · Reviewer_uCfs · 2025-11-11

**Soundness:** 3
**Presentation:** 3
**Contribution:** 3
**Rating:** 6
**Confidence:** 4

**Summary:**

This paper aims to enhance medical mathematical reasoning capabilities in large language models (LLMs) through a knowledge-guided reward framework. The proposed approach introduces two complementary mechanisms:
1. Knowledge verification reward based on explicit formula generation, utilizing an independent verification model to assess correctness.
2. Soft-hard reward mechanism that integrates (i) clinical safety thresholds as hard constraints and (ii) progressive accuracy-based rewards as soft optimization signals.

The model is trained in two stages, supervised fine-tuning (SFT) followed by reinforcement learning (RL), and evaluated on the MedCalc benchmark.

**Strengths:**

- The proposed reward components (format, knowledge, and answer rewards) are well-motivated and align with realistic clinical reasoning and decision-making processes.
- The evaluation includes a wide range of zero-shot baselines, providing a strong comparative foundation.
- The ablation studies are comprehensive and effectively demonstrate the contribution of each reward component.

Overall the proposed framework is highly effective.

**Weaknesses:**

- The selection of constants (e.g., values 2 and 3 for the hard reward) should be justified. Are these empirically tuned or theoretically grounded?
- The proposed framework is only evaluated on Qwen, raising questions about its generalizability to other LLM backbones. Other LLM backbones should be evaluated.
- The choice of five epochs for RL training should be motivated, especially if performance saturates or diverges beyond that point.
- Table numbering in the text should follow chronological order.
- In Table 1, the best results should be boldfaced for easier interpretation.
- The code is not released, which limits reproducibility and transparency.

**Questions:**

- How was the predefined set of valid formulas constructed?
- What makes this framework specific to medical reasoning? Could the same reward design not generalize to other mathematical or scientific reasoning tasks? Please clarify the domain-specific assumptions or dependencies that limit its broader applicability

---

### Note · Authors · 2025-11-30

I have read and agree with the venue's withdrawal policy on behalf of myself and my co-authors.